# A Technology-Mediated Interventional Approach to the Prevention of Metabolic Syndrome: A Systematic Review and Meta-Analysis

**DOI:** 10.3390/ijerph18020512

**Published:** 2021-01-10

**Authors:** Gaeun Kim, Ji-Soo Lee, Soo-Kyoung Lee

**Affiliations:** Nursing College, Keimyung University, 1095 Dalgubeol-daero, Dalseo-gu, Daegu 42601, Korea; eun0325@kmu.ac.kr (G.K.); jisoo20335@gmail.com (J.-S.L.)

**Keywords:** metabolic syndrome, meta-analysis, prevention and control, technology, telemedicine

## Abstract

Background: Technology-mediated interventions help overcome barriers to program delivery and spread metabolic syndrome prevention programs on a large scale. A meta-analysis was performed to evaluate the impact of these technology-mediated interventions on metabolic syndrome prevention. Methods: In this meta-analysis, from 30 January 2018, three databases were searched to evaluate interventions using techniques to propagate diet and exercise lifestyle programs for adult patients with metabolic syndrome or metabolic risk. Results: Search results found 535 citations. Of these, 18 studies met the inclusion criteria analyzed in this article. The median duration of intervention was 4 months and the follow-up period ranged from 1.5 to 30 months. The standardized mean difference (SMD) between the two groups was waist circumference −0.35 (95% CI −0.54, −0.15), triglyceride −0.14 (95% CI −0.26, −0.03), fasting blood glucose −0.31 (95% CI −0.42, −0.19), body weight −1.34 (95% CI −2.04, −0.64), and body mass index −1.36 (95% CI −2.21, −0.51). There was no publication bias in this study. Conclusion: Technology-mediated intervention improved clinically important metabolic syndrome related indicators such as excess body fat around the waist, fasting glucose, and body mass index. These interventions will play an important role in the dissemination of metabolic syndrome prevention programs.

## 1. Background

Metabolic syndrome is characterized by clinical features such as insulin resistance, abdominal obesity, hypertension, hypertriglyceridemia, high blood-sugar, and low HDL-C of which at least three were defined as concurrently occurring [1,2,3]. The prevalence of adults with metabolic syndrome is increasing worldwide [4]. There are reports that the risk of developing cardiovascular disease in subjects with metabolic syndrome is more than 2 times higher than general subjects, and the risk of developing diabetes is 3.5 to 5 times higher than that of general subjects [5].

In addition, a sedentary lifestyle, high-calorie diet, and sweet drinks are also risk factors that increase the onset of metabolic syndrome. However, regular physical activity is known to lower an individual’s risk [6,7,8,9]. Moreover, one study reported that better knowledge of one’s illness improves an individual’s lifestyle by increasing access to health care and positively affecting the treatment process and self-care [10,11].

In recent studies, the overall prevalence rate of metabolic syndrome among South Korean adults fluctuated around 28% in 2013–2015 [4]. At the same time, metabolic syndrome incurs high costs not only to the patient but also to the entire community [12]. According to a report from the Korea Centers for Disease Control and Prevention (KCDC) 65 [13], 7 of the top 10 leading causes of death are chronic diseases, including metabolic syndrome. The treatment costs amount to approximately 38 trillion won, accounting for about 80% of all medical expenses. It is suggested that the potential medical costs associated with metabolic syndrome could increase exponentially [14].

According to previous studies, the middle-aged and many young people are at risk of developing lifestyle-related diseases [15,16]. In reflection of the risk and trend of metabolic syndrome, the Korean government has implemented multiple preventive healthcare projects since 2018 [17,18,19]. The New Health Plan 2020, launched by the Department of Public Health in 2011, prioritizes the prevention of highly prevalent adult diseases such as diabetes and high blood pressure, ultimately increasing the national disability-adjusted life expectancy [14,20,21]. Essentially, the prevalence of metabolic syndrome in Koreans has been steadily increasing for 10 years [22,23,24]. 

In addressing these issues, several studies have used technology-mediated interventions to promote health in participants at risk for metabolic syndrome (MetS) [16,25,26,27,28,29]. According to a literature review related to the prevention and management of MetS, about a third of the studies were only targeting the middle-aged population, and over 60% were concerned about people already living with MetS. On the other hand, studies on young adults, a group with the highest success rate of MetS prevention, were scarce [25,30,31,32,33]. 

Thus, concerning the difficulty of substantially changing the lifestyle of the middle-aged population, it is essential to develop an effective MetS prevention program that can support the establishment of a healthy lifestyle [34,35,36]. This result implies the importance of more tailored preventive measures that aim for a particular target population [37,38,39]. 

Based on such factors, tech-mediated interventions should have a significant impact on the prevention of MetS [40,41,42,43]. To date, however, a comprehensive systematic review of these studies has not been performed. Therefore, this study does not aim to prevent metabolic syndrome through technology-mediated interventions but to evaluate the impact of these interventions on improving the metabolic profile.

## 2. Methods

### 2.1. Search Strategy and Study Selection

This systematic review and meta-analysis were done in adherence to the Preferred Reporting Items for Systematic Reviews and Meta-Analyses (PRISMA) statement guidelines [44]. We examined studies evaluating interventions that used technology to disseminate diet and exercise lifestyle programs for adult patients with MetS or metabolic risks. This study authors searched Medline, EMBASE, and the Cochrane controlled trials register (CENTRAL) from 1 January 2000, to 31 January 2018. Search terms to assess lifestyle intervention and use of technology were used, including the combination of MeSH and Emtree headings and subheadings, free-text keywords, and study design filters. The search strategy included (metabolic AND (syndrome OR risk)) AND (wearable OR app OR application OR mobile OR smartphone OR Internet OR web OR technology OR (social media) OR ((e OR m OR u OR ubiquitous OR tele) AND (health OR medicine OR nursing))). 

We manually searched reference lists of review articles, and experts in the field were contacted to include all possible studies. Studies were included in the review if they met the following criteria: (1) Population: Adults aged 18–65 years with MetS or metabolic risks, (2) intervention: Technology-based intervention, (3) comparison: No treatment, usual care, other intervention without technology, (4) outcomes: MetS-related outcome measures (waist circumference (WC), triglycerides (TG), high-density lipoprotein cholesterol (HDL), systolic blood pressure (SBP), diastolic blood pressure (DBP), fasting glucose (FG)), body weight (Body Wt), body mass index (BMI), low-density lipoprotein cholesterol (LDL), and hemoglobin A1c (HbA1c); (5) Study designs: Randomized controlled trials (RCT), (6) published in English, and (7) published in a peer-reviewed journal. Technology-based interventions included web-based programs, e-mail counselling, mobile devices such as cell phones, patient monitoring devices, personal digital assistants (PDAs), social media interventions, and other wireless devices. Studies were accepted if they used short messaging services (SMS) and more complex functionalities, such as Bluetooth technology and smartphone applications. Two authors (JSL and GK) independently screened the studies based on the inclusion criteria. If differences between reviewers persisted, a third author (SKL) resolved discrepancies by discussion until a consensus was reached.

### 2.2. Data Extraction

We extracted data from the RCTs included in the studies following the recommendations of the Cochrane Handbook for Systematic Reviews of interventions [45]. For each included study, reviewers independently extracted data including study background information (publication year, country, authors), sample-related information (eligibility, number of participants, participants’ characteristics), intervention-related information (contents, technique, duration, follow-up), comparator related information, outcome-related information (WC, TG, HDL, SBP, DBP, FG, Body Wt, BMI, LDL, HbA1c). Discrepancies were resolved through discussion.

### 2.3. Risk of Bias Assessment

The internal validity of the included studies was appraised using the Cochrane Collaboration’s tool for assessing the risk of bias in each of the domains: Selection, performance, detection, attrition, and reporting. A judgment of high, low, or unclear risk was given to the following sources of bias: Sequence generation, allocation concealment, blinding of personnel and outcome assessors, incomplete outcome data, selective outcome reporting, and other sources of bias. Unclear risk of bias was assigned when there was a lack of information or uncertainty. The authors of the included studies were contacted to clarify details about the different criteria for allocation of risk of bias and lack of clarity.

### 2.4. Statistical Analyses

Effect sizes were calculated based on the mean changes in scores of the intervention and control groups and their reported standard deviations (SDs). We extracted continuous data as means and SDs. Where change scores were not reported, pre- and post-intervention values were used to calculate the change score, and SDs were estimated as prescribed by the Cochrane handbook for systematic reviews of intervention. Effect sizes across studies were summarized for each domain using the random-effects model. Random effects models assume that the surveys are drawn from unequal populations and therefore account for the variation in the underlying effects in the estimates of uncertainty. 

When needed, subgroup and sensitivity analyses were conducted. We divided subjects into intervention contents, components, and techniques. The heterogeneity of the studies was tested using the Higgins.

*I*^2^ statistic, and significant heterogeneity was defined as 50% of the *I*^2^ value. The chi-square test and Higgins *I*^2^ were included in the forest plots. Based on the heterogeneity of the included studies, fixed or random-effects models were selected to calculate the pooled effect measures. Funnel plots for each outcome were also prepared and evaluated to assess potential publication bias. Egger’s intercepts for each outcome were also examined to determine potential publication bias. We performed all analyses using the Cochrane Collaboration software (RevMan ver. 5.3.3 The Cochrane Collaboration, 2014. Nordic Cochrane Centre, Copenhagen, Denmark) and Comprehensive Meta-Analysis version 2.2 (CMA).

## 3. Results

### 3.1. General Characteristics of the Studies

Our initial literature search yielded 535 citations, 174 of which were duplicate studies. Following the screening process, a total of 289 surveys were excluded by title and abstract. The resulting 72 publications were reviewed in full, 61 of which were excluded because they did not meet the inclusion criteria. Finally, 18 studies [46,47,48,49,50,51,52,53,54,55,56,57,58,59,60,61,62,63] were ultimately identified as relevant to our review (see Figure 1).

We analyzed 18 studies published between 2007 and 2018, and the median number of samples was 77, and the range was 22 to 1032. A total of seven studies [46,53,54,56,57,60,62] were conducted in the USA, three studies [50,51,59] in Korea, and two studies [58,63] in Japan. The remaining six studies were conducted in Brazil [47], Canada [61], Germany [52], Greece [48], Iran [49], and Taiwan [55]. Seven studies [46,47,48,49,50,51,52] were conducted on patients with MetS as the subjects, and eleven studies [53,54,55,56,57,58,59,60,61,62,63] were conducted on patients with metabolic risk. 

The median age of the participants was 49.62 years (range, 37.93–59.70). Twelve studies [48,49,50,51,52,53,55,56,57,58,60,62] included dietary and physical activity intervention, and single dietary intervention was two studies [47,63] physical activity single intervention was four studies [46,54,59,61]. The median duration of intervention was four months (range, 1.5–30 months), and the range of the follow-up periods was between 1.5 and 30 months. For web-based interventions, access to the program was unlimited, and comments or feedback were provided once to four times a month, and text was sometimes two to five times a day (see Table 1).

Most studies measured the results of the intervention at the beginning and end of the studies. Two of the studies [51,54] were based on the transtheoretical model, and there was a program that included motivation and behavior change and self-efficacy strategies based on ATP III guidelines and goal-setting theory. The techniques used in 18 interventions include web-based resources DVD and electronic video [50,53,57] telephone consultation [47,48,50,52,56,57,59,61,62,63] text message [51,59,60] e-mail [46,49,56,57,62] web-site contact [54,55,58] and online discussion [46]. The lessons and messages delivered via the technology-enabled interventions centered on educating participants on how to achieve a healthy diet and exercise to reduce the risk of metabolic risk or MetS-related signs. It also focused on symptoms and enabling behavioral changes through goal setting, self-monitoring, and logging of diet and physical activity. Video, text messages or Web-based lessons often introduce diet and physical activity concepts. In contrast, personalized or automated phones, text messages, and e-mail messages would reinforce concepts, goals, and self-monitoring behavior. In the comparison group, five out of 18 were in the no-treatment group, four were in the usual care group, and others were in the brief booklet or self-help control group.

### 3.2. Methodological Quality and Risk of Bias

Figure 2 presents an assessment of the risk of bias for these studies. Most studies have reported proper random order generation methods (risk of selection bias). Five studies reported allocation concealment, but other studies were unclear in this regard. Most studies did not report allocation concealment, but because of the nature of the study, it was not possible to blind participants to intervention allocation. Patient and practitioner blinding, and the possibility of performance bias was low or unclear in most studies. The risk of bias associated with assessor blinding and selective reporting was low in most studies. Overall, most of the RCTs were judged to raise some concerns in at least one domain, but not to be at high risk of bias.

### 3.3. Publication Bias

No significant asymmetry appeared in the inverted funnel plots of these RCTs (see Figure 3). Egger’s test also showed no potential for publication bias (*p* < 0.05). Therefore, the RCTs included in this analysis had no publication bias. However, since unpublished studies were excluded in our study, there is a risk of publication bias due to this.

### 3.4. Metabolic Syndrome-Related Outcomes

The effect of the technology-mediated MetS intervention was assessed by measuring the waist circumference (WC), triglycerides (TG), high-density lipoprotein cholesterol (HDL), systolic blood pressure (SBP), diastolic blood pressure (DBP), fasting glucose (FG), body weight (Body Wt), body mass index (BMI), low-density lipoprotein cholesterol (LDL), and hemoglobin A1c (HbA1c). 

#### 3.4.1. Waist Circumference (WC)

There were eleven studies [47,48,49,50,51,52,53,55,56] on waist circumference among studies comparing technology-mediated intervention and control. The ratio of actual variance to the total variance (*I*^2^) was 59.95%, which was analyzed as a random effect model. As a result, standardized mean differences (SMD) between the two groups were −0.35 (95% CI −0.54, −0.15), and there was a statistically significant difference (*p* < 0.001) (Figure 3). There was no publication bias (Egger’s test; *p* = 0.12) (Figure 4). 

The results of the subgroup analysis of exercise, diet, and exercise according to intervention contents, exercise, and diet intervention were SMD −0.36 (95% CI −0.58, −0.14, *p* = 0.001). However, the heterogeneity among the studies was still moderate (*I*^2^ = 64.51). The exercise alone intervention was significantly reduced (SMD −0.59, 95% CI −1.17, −0.01, *p* <0.05), but there was no statistically significant difference in diet alone intervention SMD −0.07 (95% CI −0.50, 0.37, *p* = 0.772).

#### 3.4.2. High-Density Lipoprotein Cholesterol (HDL)

There were twelve studies [46,47,48,49,50,52,53,55,56,58] on high-density lipoprotein cholesterol among studies comparing technology-mediated intervention and control. *I*^2^ was 95% which means significant heterogeneity, this has to be mentioned. As a result, SMD between the two groups was 0.77 (95% CI 0.20, 1.34, *p* < 0.01), and there was a statistically-significant difference (*p* < 0.01) (Figure 3) There was no publication bias (Egger’s test; *p* = 0.13) (Figure 4). 

The result of a subgroup analysis by intervention type (exercise, diet) as follows. The exercise alone intervention was not significantly reduced (SMD 0.64, 95% CI −0.22, 1.50, *p* = 0.15), and there was no statistically significant difference in diet alone intervention (SMD 0.08, 95% CI −0.35, 0.52, *p* = 0.71).

#### 3.4.3. Low-Density Lipoprotein Cholesterol (LDL)

There were nine studies [46,47,49,52,53,56,58] on low-density lipoprotein cholesterol among studies comparing technology-mediated intervention and control. *I*^2^ was 0.00%, which was analyzed as a fixed effect model. As a result, the SMD between the two groups was −0.24 (95% CI −0.37, −0.12), and there was a statistically significant difference (*p* < 0.01) (Figure 3). There was no publication bias (Egger’s test; *p* = 0.40) (Figure 4). 

#### 3.4.4. Triglycerides (TG)

There were eleven studies [46,47,49,50,52,53,55,56,58] on triglycerides among studies comparing technology-mediated intervention and control. *I*^2^ was 0.00%, which was analyzed as a fixed effect model. As a result, the SMD between the two groups was −0.14 (95% CI −0.26, −0.03), and there was a statistically significant difference (*p* < 0.05) (Figure 3). There was no publication bias (Egger’s test; *p* = 0.62) (Figure 4). 

#### 3.4.5. Systolic Blood Pressure (SBP)

There were eleven studies [48,49,50,52,53,55,56,58,59] on systolic blood pressure among studies comparing technology-mediated intervention and control. *I*^2^ was 43.70%, which was analyzed as a fixed effect model. As a result, the SMD between the two groups was −0.25 (95% CI −0.37, −0.14), and there was a statistically significant difference (*p* < 0.001) (Figure 3). There was no publication bias (Egger’s test; *p* = 0.06) (Figure 4). 

#### 3.4.6. Diastolic Blood Pressure (DBP)

There were eleven studies [48,49,50,52,53,55,56,58,59] on diastolic blood pressure among studies comparing technology-mediated intervention and control. *I*^2^ was 57.60%, which was analyzed as a random effect model. As a result, SMD between the two groups was −0.32 (95% CI −0.51, −0.13), and there was a statistically significant difference (*p* = 0.001) (Figure 3). There was no publication bias (Egger’s test; *p* = 0.38) (Figure 4). 

The results of the subgroup analysis of exercise, diet, and exercise according to intervention contents, exercise, and diet intervention was SMD −0.28 (95% CI −0.46, −0.10, *p* < 0.01), but the heterogeneity among the studies was still moderate (*I*^2^ = 52.36). There was a statistically significant difference in exercise alone intervention (SMD −0.95, 95% CI −1.54, −0.35, *p* < 0.01).

#### 3.4.7. Fasting Glucose (FG)

There were eight studies [47,48,49,50,55,56,58] on fasting glucose among studies comparing technology-mediated intervention and control. *I*^2^ was 0.00%, which was analyzed as a fixed effect model. As a result, the SMD between the two groups was −0.31 (95% CI −0.42, −0.19), and there was a statistically significant difference (*p* < 0.001) (Figure 3). There was no publication bias (Egger’s test; *p* = 0.12) (Figure 4).

### 3.5. Other Outcomes

#### 3.5.1. Body Mass Index (BMI)

There were ten studies [47,48,49,52,53,54,57,58] on body mass index among studies comparing technology-mediated intervention and control. *I*^2^ was 97.32%, which was analyzed as a random effect model. As a result, the SMD between the two groups was –1.36 (95% CI −2.21, −0.51), and there was a statistically-significant difference (*p* < 0.01) (Figure 3). There was no publication bias (Egger’s test; *p* = 0.06) (Figure 4). 

The results of the subgroup analysis of exercise, diet, and exercise according to intervention contents, exercise and diet intervention was SMD = −1.70 (95% CI −2.78, −0.61, *p* < 0.01), but the heterogeneity among the studies was still high (*I*^2^ = 98.00). The exercise alone intervention was not significantly reduced (SMD −0.39, 95% CI −1.10, 0.31, *p* = 0.28), and there was no statistically significant difference in diet alone intervention (SMD −0.05 (95% CI −0.49, 0.38), *p* = 0.81).

#### 3.5.2. Body weight (Body Wt)

There were sixteen studies [47,49,51,52,54,55,56,57,58,59,60,62] on bodyweight among studies comparing technology-mediated intervention and control. *I*^2^ was 98.31%, which was analyzed as a random effect model. As a result, the SMD between the two groups was –1.34 (95% CI −2.04, −0.64), and there was a statistically-significant difference (*p* < 0.001) (Figure 3). There was no publication bias (Egger’s test; *p* = 0.83) (Figure 4). 

The results of subgroup analysis of exercise, diet, and exercise according to intervention contents, exercise and diet intervention was SMD = −1.62 (95% CI −2.41, −0.83, *p* < 0.001), but the heterogeneity among the studies was still high (*I*^2^ = 98.57). The exercise alone intervention was not significantly reduced (SMD −0.20, 95% CI −0.64, 0.24, *p* = 0.38), and there was no statistically significant difference in diet alone intervention (SMD −0.05, 95% CI −0.49, 0.39, *p* = 0.82).

#### 3.5.3. Hemoglobin A1c (HbA1c)

There was one study [63] on the HbA1c index among studies comparing technology-mediated intervention and control, and SMD between the two groups was –0.53 (95% CI −0.11, 1.16), and there was no statistically-significant difference (*p* = 0.103) (Figure 3).

### 3.6. Sensitivity Analysis

The results showed that the pooled standardized mean difference was ∣0.14∣ (95% CI, –0.28, –0.01) ~∣1.40∣ (95% CI, –2.32, –0.49) for the random-effects model on prevention of MetS. We excluded individual studies from the sensitivity analysis but excluding them did not affect the initial effect size estimates (see Figure 5).

## 4. Discussion

This study attempted to grasp the current status of existing studies on the effect of technological interventions for MetS prevention on MetS prevention and systematically investigate its effects. A total of 18 studies over the past eight years (2000–2018) were analyzed to select the effectiveness of technology-mediated interventions for MetS prevention and to apply exclusion criteria. As for the study design, a total of 18 studies were randomized controlled studies, and all studies reported appropriate methods of random sequence generation (risk of selection bias). Assignment concealment was reported in five studies. However, most studies did not report allocation concealment, so the risk of bias was rated as “low” Overall, most RCTs are classified as having a low risk of bias, indicating a high level of research.

All interventions in the experimental group were technical intervention programs followed by eight web-based resources, 2 DVDs and electronic videos, one video conference, and eight telephone consultations. In looking at the analyzed papers, most studies showed a positive intervention effect. This is due to the rapid technological development of modern society, which makes it possible to access and use information resources in ubiquitous ways through the Internet and various mobile devices in daily life. As more people use technology and the Internet to search, technology-mediated interventions have great potential. They are thought to have a positive impact on the treatment and prevention of MetS [64,65,66].

Technology-mediated interventions can address these limitations and inconveniences while delivering reliable educational materials to many people at a convenient time and place [67]. People can receive consistently and the same intervention regardless of healthcare providers’ level of expertise. In considering the high information needs of MetS patients and the significance of adequate intervention, well-developed technology-supported interventions can benefit many patients with MetS. In general, MetS can be prevented by changing diet and lifestyle, but consistently maintaining it can be difficult. Technology-mediated interventions must improve personal-level physical activity and good eating habits and encourage users to engage in ongoing health care by recommending and encouraging the use of appropriate mobile apps and wearable devices. The results of this study are consistent with results from previous studies that revealed the importance of technology-mediated interventions with no time and space limitations for a sustained increase in physical activity [16,27,40]. In this way, technology-mediated intervention is not only excellent in accessibility but also technology-mediated interventions might be more cost-effective and easier to scale up than traditional educational programs. However, although technology-mediated arbitration is on the rise, not all of them are accessible. There are still many environments in which the internet is not accessible due to the internet environment or servers; therefore, these potential limitations must be considered.

In this study, there are a total of 18 technical interventions for patients with MetS. The effectiveness of the intervention program includes indicators related to body measurements such as body mass index, weight, and blood pressure as well as hematologic indicators such as blood sugar, high- or low-density cholesterol, triglycerides, insulin, and body fat distribution. There was an emphasis on measuring physical indicators such as composition indicators, and no assessments of the impact of psychosocial indicators such as emotion, perception, and quality of life have been reported. Therefore, it is judged to be very important to evaluate multidimensional effects in terms of mental and social aspects. 

Although this meta-analysis confirmed the homogeneity of the study, technology-mediated interventions were found to be effective in reducing waist circumference, fasting blood sugar, and body mass index, with statistically significant results. Abdominal obesity increases the risk of developing chronic diseases such as diabetes, hypertension, dyslipidemia, and fatty liver [16,29,40,49]. Abdominal obesity is known to have a greater risk of MetS due to not only simple abdominal obesity, but also other clinical features of MetS such as blood pressure, high triglycerides, hypoglycemic lipoprotein cholesterol, and blood sugar [7,36,68]. The effectiveness of technology-mediated interventions for MetS is quite significant and meaningful.

Looking at the results of this study, it is recommended to provide at least 6 weeks of intervention and exercise to reduce waist circumference as a technique-mediated intervention. This is consistent with previous studies that reported that four months of intervention to address lifestyle issues was sufficient [47,48,53,56]. However, the sub-analysis results appear to have low intensity and publication bias due to the small number of studies, so attention should be paid to interpretation. In addition, it is considered to be of considerable significance that the effect size of technology mediated intervention is small. Still, meaningful analysis results were confirmed by performing meta-analysis without publication bias through the fixed-effect analysis method. 

Additionally, as evidenced by our finding that 4 of 18 intervention arms [55,58] reported sustained weight loss outcomes at least one-year post-intervention, the waist size reduction achieved through technology-mediated interventions may be sustainable. This finding supports the claim that technology-mediated interventions are an effective way to prevent the development of MetS.

From a clinical standpoint, we may soon live in a world where provider referrals to technology-mediated interventions to promote lifestyle and behavior change are commonplace. This market is filled with products whose development often, theoretical underpinnings, and an appropriate evidence base to support use [16,69,70,71]. The meta-analysis of this study also demonstrates that quality technology-mediated interventions are effective in preventing MetS, so more steps should be taken to facilitate their use in clinical practice. This is the same opinion as Lee’s16 research results, and it is a primary measure to realize effective technology-mediated interventions. 

Meta-analysis is useful in that it can be generalized by quantitatively integrated analysis of existing studies that have individually reported the effectiveness of studies and provides a reasonable basis for making clinical decisions. However, in this study the number of selected papers was small due to the limitations of the research papers with qualitative validity and the random distribution method when selecting papers. In addition, since the heterogeneity of the paper included in the analysis is generally high, subgroup analysis and sensitivity analysis were performed to solve this issue. And since unpublished studies were excluded in our study, there is a risk of publication bias due to this. There was also a limit to the selection of papers in English only. So, the study results must be accepted with caution. Also, the field of technology-based arbitration is changing rapidly. Therefore, in keeping with these changes, the next study suggests meta-analysis as the most recent study. A follow-up study that repeatedly verifies the results of this study is recommended.

## 5. Recommendations for Future Research

This study clearly revealed that metabolic syndrome-related technology-mediated interventions have ample opportunities to promote lifestyle changes. In particular, technology-mediated interventions helped lower the circumference of the abdomen, blood sugar, and body mass index. Currently, this type of research is the cornerstone to establish a wider range. Hence, future studies should apply a rigorous study design and repeat studies to further investigate the use of technology-mediated techniques.

## 6. Conclusions

Our meta-analysis demonstrated that technology-mediated MetS prevention interventions are effective in improving all components of the MetS and have statistically significant results. These results suggest that technology-mediated interventions could be an alternative to in-person MetS prevention programs. The option of using technology-mediated delivery can potentially overcome barriers of access and allow for expanded dissemination of such interventions.

## Figures and Tables

**Figure 1 ijerph-18-00512-f001:**
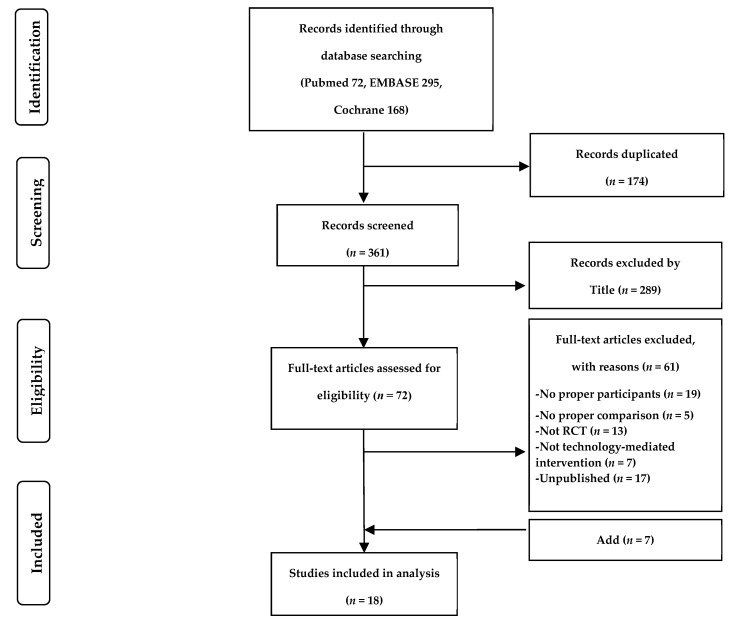
Flowchart of the process for selecting studies for the systematic review.

**Figure 2 ijerph-18-00512-f002:**
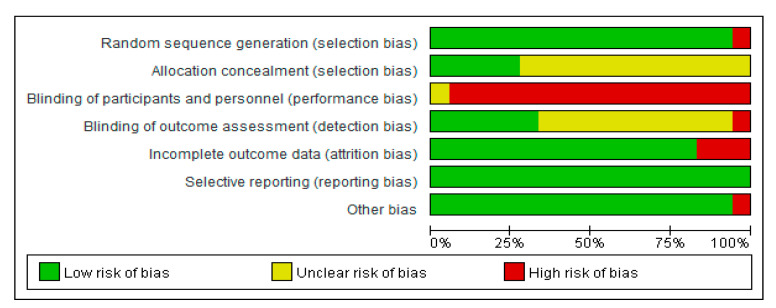
Assessment of risk of bias in included studies.

**Figure 3 ijerph-18-00512-f003:**
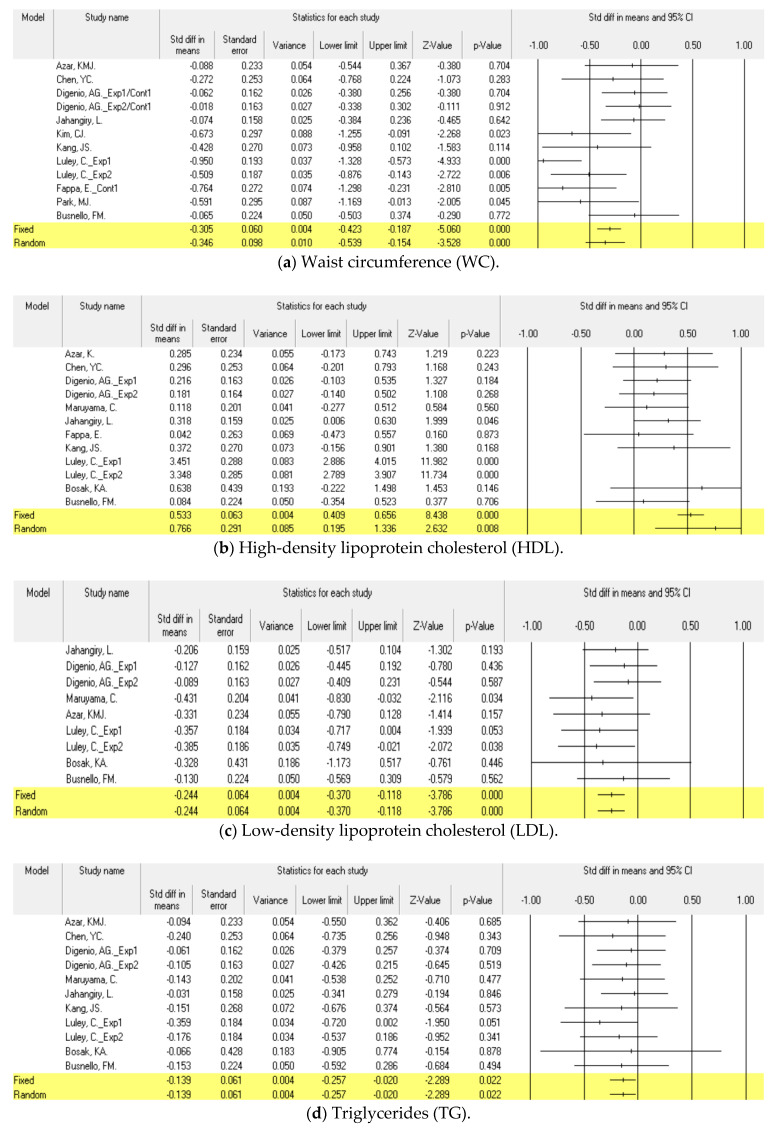
Forest plot of the meta-analysis for technology-mediated intervention on prevention of metabolic syndrome. Each study is identified by first author. The individual effect sizes are identified as “standardized mean difference” with lower and upper limits (95% confidence intervals). The overall summary effect size of the meta-analysis is noted as a diamond on the bottom line.

**Figure 4 ijerph-18-00512-f004:**
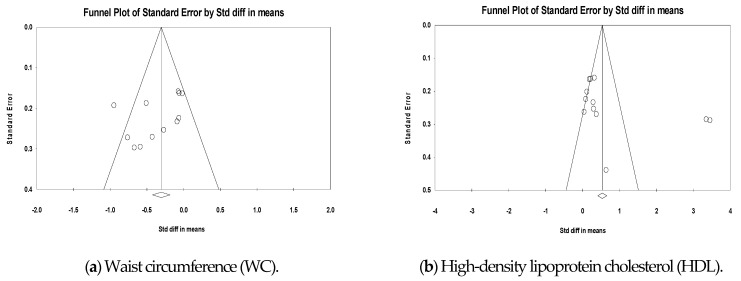
Funnel plot of the effects of the technology-mediated MetS intervention. (**a**) Waist circumference (WC), (**b**) High-density lipoprotein cholesterol (HDL), (**c**) Low-density lipoprotein cholesterol (LDL), (**d**) Triglycerides (TG), (**e**) Systolic blood pressure (SBP), (**f**) Diastolic blood pressure (DBP), (**g**) Body mass index (BMI), (**h**) Body weight (Body Wt), (**i**) Fasting glucose (FG).

**Figure 5 ijerph-18-00512-f005:**
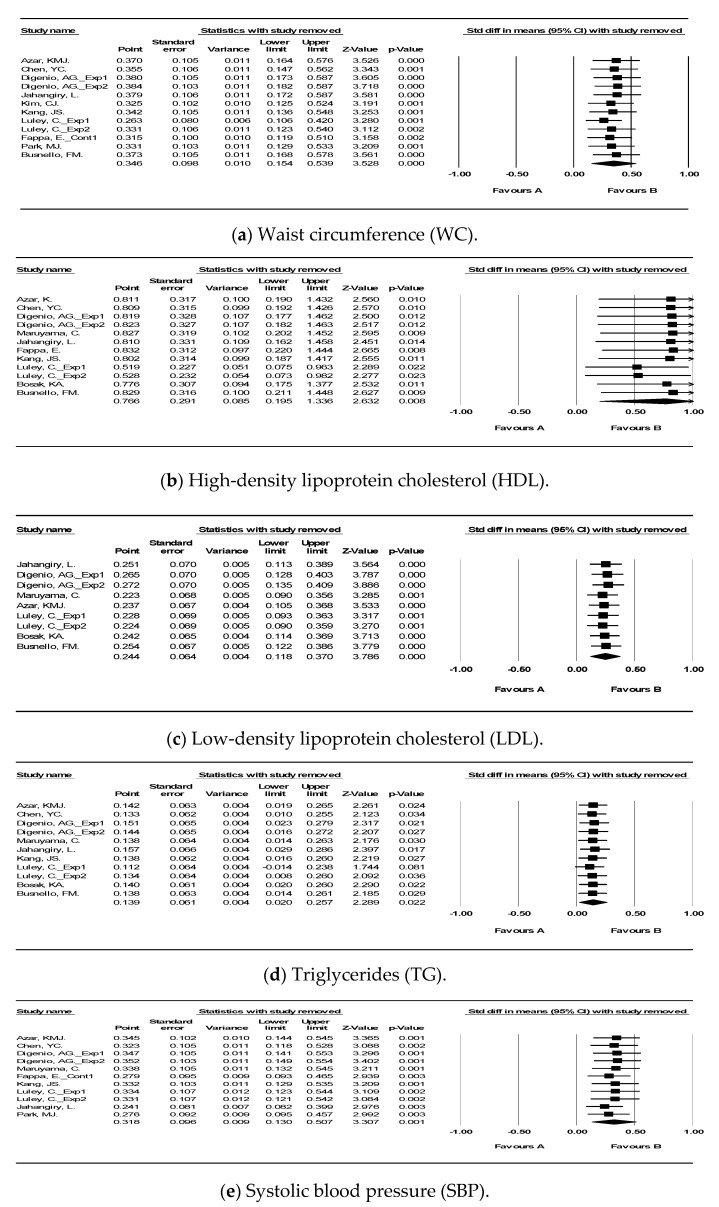
Forest plot after sensitivity analyses of the meta-analysis for the technology-mediated MetS intervention. (**a**) Waist circumference (WC), (**b**) High-density lipoprotein cholesterol (HDL), (**c**) Low-density lipoprotein cholesterol (LDL), (**d**) Triglycerides (TG), (**e**) Systolic blood pressure (SBP), (**f**) Diastolic blood pressure (DBP), (**g**) Body mass index (BMI), (**h**) Body weight (Body Wt), (**i**) Fasting glucose (FG).

**Table 1 ijerph-18-00512-t001:** Main characteristics of included studies.

First Author, Public Year	Ref No	Country	Subject		Criteria	Intervention				Control
			N (M:F)	Age (Years) Mean (SD)	Inclusion	Intervention (Contents)	Intervention (Components and Technique)	Duration (Months)	f/u (Months)	
Bosak, KA., 2010	[46]	USA	22 (16:6)	50.94 (7.81)	Adults with metabolic syndrome	Exercise	Web-based education programme (Internet physical activity intervention) based on the ATP III guidelines Self-efficacy strategies e-mail feedback, Quiz (week 5, week 6) Discussion via electronic discussion board	1.5	1.5	Usual care
Busnello, FM., 2011	[47]	Brazil	80 (23:57)	58.50 (8.50)	Patients with metabolic syndrome	Diet	Individual standard diet and a “Manual of Nutritional Guidelines for Patients with Metabolic Syndrome” Telephone counselling Different printed material about nutrition guidelines	4	4	Usual care (nutritional guidance)
Fappa, E., 2012 (1) Cont 1 (2) Cont 2	[48]	Greece	87 (50:37)	49.00 (11.80)	Patients with metabolic syndrome	Exercise, Diet	Based on the goal setting theory Motivational and behavioral strategies -Telephone counselling intervention (1~2 times/month, total 7 times)	6	6	Cont. 1: Usual care Cont. 2: Face-to face counselling (1~2 times/month, total 7 times)
Jahangiry, L., 2015	[49]	Iran	160 (106:54)	44.05 (10.05)	Patients with metabolic syndrome	Exercise, Diet	Interactive web-based programme lifestyle intervention (the Healthy Heart Profile: Education, diet information, estimation of FSR, personal health records), ∙e-mail and encouraged	6	6	e-mail
Kang, JS., 2014	[50]	Korea	56 (46:10)	37.93 (10.13)	Adults with metabolic syndrome	Exercise, Diet	Web-based health promotion programme (audio-video clips on diet and exercise using the internet) One of researchers contacted the participants by telephone to reinforcement No offline coaching	2	2	Usual care (Brief booklet)
Kim, CJ., 2015	[51]	Korea	48 (48:0)	39.63 (7.27)	Male workers with metabolic syndrome	Exercise, Diet	Internet-based Best Exercise Super Trainer (BEST program: Multi-component WBI incorporating physical activity/weight control, personal counselling) lifestyle intervention Based on transtheoretical model (TTM) Internet-based online counselling (1 times/week) Short mobile text messages (SMS)	4	4	Usual care (Brief booklet)+ SMS
Luley, C., 2014 (1) Exp 1 (2) Exp 2	[52]	Germany	178 (105:73)	50.25 (7.96)	Patients with metabolic syndrome	Exercise, Diet	Mobile technology based lifestyle intervention (nutrition and physical activity): Both intervention groups were issued accelerometers (Aipermotion 440), which measured physical activity, recorded daily weight and calorie intake, and transmitted these data to a central server for use by patient carers. +Exp 1: Active Body Control (ABC) lifestyle program, information and motivation by letters (1 times/week) +Exp 2: 4 sigma coaching intervention, Telephone counselling (1 times/month)	12	4,8,12	Usual care
Azar, KMJ., 2016	[53]	USA	74 (30:44)	59.70 (11.20)	Adults with cardiometabolic risk (1) BMI ≥ 35 kg/m^2^ and prediabetes, previous gestational diabetes and/or metabolic syndrome (2) BMI ≥ 30 kg/m^2^ and type 2 diabetes and/or cardiovascular disease	Exercise, Diet	Electronic CardioMetabolic Program (eCMP, web-based comprehensive program) The delivery of evidence-based curricula using online tools Pre-recorded didactic videos presented by physicians, nutritionists, exercise physiologists, and lifestyle coaches. -A comprehensive online platform and participant portal for hosting programme materials (e.g., homework assignments, didactic videos, and calendars) Face-to-face group meetings (1 times/week) via web-based video conferencing Mobile monitoring devices: Self-monitoring, bio-feedback, remote data capture (wireless body scale (Fitbit and Withings Smart Scale), pedometer) Coach-led virtual small groups via real-time, encrypted, web-based videoconferencing (4 times/month) Coach-led in-person sessions (periodic 7 sessions)	6	3, 6	No treatment
Carr, LJ., 2008	[54]	USA	32 (6:26)	45.90 (2.75)	Adults with metabolic syndrome risk Sedentary overweight (BMI ≥ 25.0 kg/m^2^)	Exercise	The ALED-I (active living every day internet-delivered) theory-based behavior change programme (based on transtheoretical model (TTM)) Website content and functionality (Blair et al., 2001): Interactive activities and behavior modification strategies	4	4	No treatment
Chen, YC., 2013	[55]	Taiwan	63 (0:63)	43.80 (9.07)	Full time career women with metabolic syndrome risk	Exercise, Diet	Internet-based Health Management Platform (HMP) program The Internet platform included a health examination database, nutrition management system, and exercise management system. Participants were able to log into the system with individual passwords to check personal test data and upload personal dietary and exercise records. Health management experts also provided nutrition and exercise recommendations and advice according to these records.	1.5	1.5	No treatment
Digenio, AG., 2009 (1) Exp 1/Cont 1 (2) Exp 1/Cont 2 (3) Exp 1/Cont 3 (4) Exp 2/Cont 1 (5) Exp 2/Cont 2 (6) Exp 2/Cont 3	[56]	USA	376 (50:326)	43.79 (9.51)	Patients with metabolic syndrome risk 30 kg/m^2^ < BMI < 40 kg/m^2^	Exercise, Diet	Lifestyle modification counselling -Exp 1: High frequency telephone counselling (2~4 times/month) -Exp 2: High frequency E-mail counselling (2~4 times/month)	6	0.5, 1, 3, 6	-Cont 1: No treatment -Cont 2: High frequency face to face counselling (2~4 times/month) -Cont 3: Low frequency fact to face counselling (1 times/month)
Ma, J., 2013 (1) Exp 1 (2) Exp 2	[57]	USA	241 (129:112)	52.90 (10.60)	Patients with metabolic syndrome risk BMI ≥ 25 kg/m^2^ fasting glucose level 100–125 mg/dL	Exercise, Diet	∙Lifestyle intervention -Exp 1: A coach-led, group delivered intervention (group Lifestyle Balance, GLB, 12 session), web-based education, e-mail (or telephone) motivational message -Exp 2: A self-directed home-based DVD intervention	15 (Intensive intervention 3, maintenance 12)	15	Usual care
Maruyama, C., 2010	[58]	Japan	101 (101:0)	39.49 (7.89)	Patients with metabolic syndrome risk	Exercise, Diet	Life Style Modification web-based counselling programme (Physical Activity and Nutrition), counselling (1 times/month), web site advice (1 times/month)	4	4	No treatment
Park, MJ., 2009	[59]	Korea	49 (26:23)	53.8 (8.89)	Patients with metabolic syndrome risk (hypertension and obesity) BP > 120/80 mmHg BMI > 23 kg/m^2^	Exercise	Cellular telephone and Internet-based individual intervention Web-based diary through the internet or by cellular telephones (weekly) Internet recommendation and SMS message	2	2	No treatment
Patrick, K., 2009	[60]	USA	65 (13:52)	44.90 (7.70)	Patients with metabolic syndrome risk Overweight (BMI > 25–39.9 kg/m^2^)	Exercise, Diet	Text Message-based intervention (weight loss program) Counselling sessions & web site advice (1 times/month) SMS or MMS 2~5 times/day	4	4	Usual care (Printed educational materials)
Petrella, R., 2014	[61]	Canada	149 (38:111)	57.83 (9.10)	Patients with ≥ 2 metabolic syndrome risk	Exercise	Mobile health intervention (Individualized exercise prescription) Technology kit (telephone with anywhere health monitoring application, Bluetooth™ enabled blood pressure monitor, a glucometer, and a pedometer) for home monitoring of biometrics and physical activity	12	3, 6, 12	Individualized active exercise prescription
Svetkey, LP., 2008 (1) Exp 1 (2) Exp 2	[62]	USA	1032 (378:654)	55.60 (8.70)	Patients with metabolic syndrome risk BMI 25–45 kg/m^2^ Taking medication for hypertension, dyslipidemia, or both	Exercise, Diet	Weight loss maintenance interventions -Exp 1: Interactive technology-based intervention (monthly), Web site education, e-mail, telephone (2 times/month) -Exp 2: Personal contact (1 hrs/month), telephone (5–15 min/month)	12, 30	12, 30	No treatment
Ueki, K., 2009	[63]	Japan	52 (22:19)	55.37 (11.64)	Patients with metabolic syndrome risk	Diet	Information Communication Technology (ICT) method Using sensors attached to the BP monitor, scale, and pedometer, the data were transmitted via the Internet or telephone circuitry from a telemetric information terminal Nutritional guidance using Telemetric-communication technology (e-mail or fax)	3	3	Face-to face guidance

-ef: Reference; No: Number, F: Female; M: Male—Units of measurement: BP: mmHg, BMI: kg/m^2^, Fasting glucose: mg/dL, Waist Circumference: Centimeters.

## Data Availability

Data available in a publicly accessible repository The data presented in this study are openly available in Table 1.

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
