# Peer review of "A Technology-Mediated Interventional Approach to the Prevention of Metabolic Syndrome: A Systematic Review and Meta-Analysis"

_ijerph, 2021, doi:10.3390/ijerph18020512_

Round 1

Reviewer 1 Report

The title of this paper is “A technology-mediated interventional approach to the prevention of metabolic syndrome: A systematic review and meta-analysis” and throughout the manuscript the authors emphasize MetS prevention; however, nearly half of the papers (7 out of 18) included in the systematic review are about MetS treatment. This lack of a coherent and clear message regarding the manuscript’s purpose are consistent with other systematic errors in the attribution of references, citing of statistics, labelling of tables and description of the results.

Major issues

Much of the manuscript, with the exception of the methods, is written with very poor English grammar to the extent that the meaning of sentences and paragraphs is lost. A few examples are listed here

  • Line 17, sentence beginning “Of these,…”
  • Line 41, sentence beginning “Moreover, one study…”
  • Line 55, sentence beginning “In addition, as a result…”. Note that households cannot develop MetS.
  • Line 148, sentence beginning “We added seven….”
  • Line 162, sentence beginning “For web-based interventions…”
  • Line 166, sentence beginning “Two of the studies…”
  • Line 346, sentence beginning “This study…”
  • Line 347, sentence beginning “A total of 18….”
  • Line 371, sentence beginning “Technology-mediated interventions…”
  • Line 383, sentence beginning “Therefore, it is judged…”
  • Line 407, sentence beginning “This market is filled…” Also, the cited references do not appear to be relevant to this statement.

This is a paper focused on MetS, but the criteria for its diagnosis are incorrect (hypoproteinemia, lipoprotein cholesterolemia) (Line 31)

Citing statistics – the prevalence of MetS in Korea is given as 28.8% (line 33), 20% (line 45), stable…steadily increasing (line 63-64)

Incorrect citations – this reviewer did not have time to review all of the citations for accuracy. However, the following are noted:

  • “…metabolic syndrome is the leading cause of death in many developed countries.” Neither of the cited references make this claim. In any case, reference 5 (a duplicate of reference 2) is an opinion piece published in a non-peer reviewed forum and does not provide references for the statistics it cites.

Several sentences in the introduction focus on MetS in young people, to the extent that the reader expects the paper is going to analyse articles written about that topic, but in fact all ages are included. Thus, the rationale for doing the work is unclear.

The inclusion criteria for technology-based interventions is quite broad (Lines 101-105) yet Figure 1 indicates that papers were excluded if they did not include “web-based interventions”. It is unclear what the “Not published” exclusion means, since presumably all the results were published if they were found in the cited databases.

Results section 3.2 – descriptions of ML algorithms in the results does not make sense, nor is it clear what models are referred to.

Results section 3.4 – it is unclear what this paragraph means. Also, the table heading for Table 2 is the same as for Table 1. Sub-sections of section 3.4 – each of these paragraphs is written in identical format and therefore each of them has the same grammatical errors. This comment also applies to section 3.5. Section 3.5.2 is about body weight but then discusses body mass index.

Figure legends repeat the same inaccuracies as the text, e.g. MetS prevention; web-based interventions

Some of the Discussion is merely a repetition of stating the results e.g. paragraph beginning on line 355.

The discussion implies cause and effect of the “rapid technological development of modern society” and the study outcomes without presenting evidence of such causation. Later in the same paragraph, references 61-63 are cited to show that technology can aid in treatment and prevention of MetS; however, none of these papers is about MetS. Similarly, references 16, 28 and 41 do not show evidence of sustained physical activity for people with MetS after technology-mediated interventions.

Line 365 – what limitations and inconveniences are referred to?

Line 393 – the authors state that interventions of at least 6 weeks are recommended but have not shown any data to support this conclusion.

Section 6 is not relevant to this manuscript.

Minor issues

The in-text citation formatting is inconsistent.

Table 1 – it is extremely difficult to tell where one intervention ends and the next one begins (column headed Intervention Components & Techniques)

Line 380 – the words “Body fat distribution” appear as a sentence in the middle of the line

Many of the citations are incomplete, do not provide correct authorship or have other issues with formatting.

Author Response

We have attached the file.

Reviewer 2 Report

Abstract

Authors should include the latest date search. This lit search concluded in January 2018. Kindly update and add other relevant studies. Eg:Toro-Ramos et al 2017 (Effectiveness of a Smartphone Application for the Management of Metabolic Syndrome Components Focusing on Weight Loss: A Preliminary Study.

Toro-Ramos T, Lee DH, Kim Y, Michaelides A, Oh TJ, Kim KM, Jang HC, Lim S.Metab Syndr Relat Disord. 2017 Nov;15(9):465-473)

Line 122-123: Authors need to state that ‘original authors of the included studies were contacted to clarify details about the different criteria for allocation of risk of bias and unclear….’

Line 148: Authors have added seven studies as snowballs, according to me ‘snowball sampling’ involves convenience sampling and has significant limitations, please clarify why this was used in a systematic review

Line 191: I agree that ‘allocation concealment’ in such studies would be challenging and hence would atleast allocate a ‘high risk’ to those where this was not possible/ not done and ‘unclear’ to those where authors were contacted and did not respond. I hence do not agree with the author’s judgement of’ Overall most RCTs were classified as having a low-risk bias indicating high quality studies’. I would use GRADE further as most of these would be downgraded and satisfy criteria for ‘low quality’ studies. Kindly clarify. Same issue to be addressed in Line 352-53 of discussion.

Para 3.3: Please clarify the exclusion of ‘unpublished studies’ and therefore this publication bias paragraph will need to be revised.

Line 237 and 238: Please rewrite this statement to correctly reflect the results

Line 364: Something needs to be added around ensuring follow ups, navigating the healthcare system, communication with healthcare providers, and shared decision making.

Methods: Need updated literature search and to update study as per the new search. Need a GRADE table for summary of findings to look at the evidence in totality. I suspect the evidence will be down graded.

Discussion: Authors need to include a comment somewhere if the intervention period of 4 months is sufficient for intervention around a chronic lifestyle related issue such as MetS.

Line 393: suggests providing… This line needs to be improvised

Line 417: Further research is needed to confirm these results is better instead of “It is necessary….research.’

These recent articles may provide additional points for discussion

  1. Buss VH, Leesong S, Barr M, Varnfield M, Harris M. Primary Prevention of Cardiovascular Disease and Type 2 Diabetes Mellitus Using Mobile Health Technology: Systematic Review of the Literature. J Med Internet Res. 2020 Oct 29;22(10):e21159.
  2. Arens JH, Hauth W, Weissmann J. Novel App- and Web-Supported Diabetes Prevention Program to Promote Weight Reduction, Physical Activity, and a Healthier Lifestyle: Observation of the Clinical Application. J Diabetes Sci Technol. 2018 Jul; 12(4):831-838.

Minor:

Please review spacing between words and numbers throughout the manuscript (highlighted in yellow)

Line 46: No bracket for reference number 4.

Line 55: No bracket for ref number 16

Line 100: Authors have included articles published only in English, so this should be acknowledged as a limitation

Line 111: No bracket for the reference number 46.

Line 116: two full stops

Line 135: should be I2 instead of I2, please correct throughout the manuscript.

Line 148: space needed between studies and 3

Please use word instead of numbers for numbers below 10 as convention. Eg: Line 160: should be ’two studies’ and ‘four studies’

Line 177: Again use five instead of 5 and four instead of 4

Line 178: and others were ‘in’ the brief booklet or self-help control group

Line 180: What are ML algorithms? This abbreviation has not been expanded before.

Line 228: No bracket between studies and 31,32 etc

Line 241: Lack of bracket between studies and 31-33

Line 242: I2 was 95% which means significant heterogeneity, this has to be mentioned

Figure 1:

Under Identification: ‘Cochrane’ is misspelt as Cochrnae

Under Eligibility: the articles which were not published were excluded, could this result in under reporting of publication bias? Please clarify. Were these abstracts only? Medline, EMBASE and CENTRAL as per my experience include mostly published articles

Author Response

We have attached the file.

Reviewer 3 Report

This paper systematically reviewed the use of technology-mediated interventions for preventing metabolic syndrome. The use of technology-mediated interventions is likely to become increasingly common, given the increasing use and access to technology. Thus, this topic may be of high importance, particularly for other scientists and researchers. The paper was well-organized, but minor changes to grammar/sentence structure would improve the paper. Overall, this systematic review was informative and the methods for finding/summarizing the studies were rigorous.

Below are my comments/suggestions for the authors:

Lines 37-38: “Metabolic syndrome is a leading cause of death in many developed countries.” This statement seems a bit too strong, as this statement could be easily disputed. Perhaps soften by stating that  “metabolic syndrome could become a leading cause of death” or “metabolic syndrome is among the leading causes of death.

Lines 44-46: “In recent studies, the overall prevalence rate of metabolic syndrome among South Korean adults fluctuated around 20% in 2013–2015, which implies that nearly one in five adults is at risk of developing metabolic syndrome in their life cycle.” The second half of sentence is redundant. Delete. 

Line 52: Do you mean to say middle-aged and young adults are at an increased risk?

Line 55: Reword sentence beginning with “In addition, as a result of analyzing the relationship between”

Lines 63-65: This sentence does not make sense. Is the prevalence stable or increasing?

Line 107: Do you mean that a fourth reviewer helped settle any ?

Methods:

Very nicely written methods section.

Some typos on page 5: centred, focussed

Results:

Table 1 is very comprehensive and informative. Very nicely done.

In the tables/text, please indicate units of measurement. (For example, centimeters or inches for waist circumference).

Lines 237-238: “SMD -0.59” should be included within the brackets.

Figure 5: I am not clear on what the difference is between Figures 3 and 5. Could you please make it more clear what was assessed in the sensitivity analyses? Also, please indicate what “Favours A” and “Favours B” mean.

Table 2: This table confuses me. Are these means across all included studies? What is the p-value for? Add a more informative title and describe the statistical methods in the methods section. Also, add units for each sub-item.

Discussion:

The results have some important implications for public health/clinical practice. Technology-mediated interventions might be more cost-effective and easier to scale up than traditional educational programs, for example.  I think it would be good to discuss the benefits (and limitations) of technology-mediated interventions.

Lines 349-351: Weren’t all of the included studies RCTs?

Lines 360-362: While it is true that access to the internet is increasing, there are still some who do not have access. Could you comment on this as a potential limitation of the technology-mediated interventions? I imagine this would be particularly important, if these types of interventions begin to utilize apps (rather than web pages), which may further limit accessibility.

Line 380: “body fat distribution” Not a complete sentence.

Lines 385-387: “technology-mediated interventions were found to be effective in reducing waist circumference, fasting blood sugar, and body mass index, with statistically significant results.” Your results actually showed that technology-mediated interventions improved all of the components of MetS. Maybe clarify this sentence…

Line 406: Add the word “from” so that it reads “From a clinical standpoint…"

Other minor comments:

Some of the language is too casual. Examples: “Lee’s study” (line 53) “we wanted to determine” (line 78)  “It is necessary to repeat this research through research.” (line 417)

Grammar and sentence structure could be improved, particularly in the discussion.

Author Response

We have attached the file.

Reviewer 4 Report

The authors have developed an interesting systematic review and meta-analysis focusing on to demonstrate how the technology-mediated interventions can improve the cardiometabolic profile in the population with high metabolic risk.

The study is okay building, with a solid design and a firm methodology. However, there are some considerations that the authors must to address to improve the quality and the understanding of this paper.

Keywords. Please, the authors should use all MeSH terms when it is possible. I recommend to check the following terms: a) prevention: I recommend to replace for “prevention and control”.

Background. A total of 44 references were used in the background and of these, 14 references (32%) are more than five years old. Please, this research deserves those old citations have a representation of less than 20%. I recommend checking this aspect to update the introduction.

Objectives: To my knowledge, this research does not aim to prevent metabolic syndrome through technology-mediated interventions, but to evaluate the impact of these interventions on improving the metabolic profile.

Material and methods.  In the section on assessing the risk of bias, I miss the assessment of the quality of the evidence with the GradePro tool. The quality of the evidence is essential to know to what extent we can trust that the estimate of the effect is correct, that is, how much confidence we can attribute to the magnitude of the effect on the recommendations of the intervention. Therefore, I recommend to include two tables: the summary of findings and the evidence profile grouping the studies for metabolic syndrome and metabolic risk.

Results. In lines number 180 and 182 some acronyms should be explained: ML, RF, SVM…

In Figures 3 and 5, I recommend including the I2 value for each forest plot. In this way, these figures will be better understood.

Discussion. I consider that the limitations of the study have been briefly addressed when the meta-analysis has shown that for 3 outcome variables (HDL cholesterol, BMI, Bodyweight) they showed very high heterogeneity. Do the authors believe that in addition to the sensitivity study, a meta-regression would be indicated

Author Response

We have attached the file.

Round 2

Reviewer 1 Report

Major issues

Lines 32 and 37 – the authors still include opinion pieces as references for statistical data, which should be discouraged (references 2 and 5). Reference 6 is incorrectly cited in the reference list as having authors Who, J. and Consultation, F.E. Nowhere in that report is it stated that MetS is a leading cause of death. (The metabolic syndrome is mentioned once, on page 57 of the report, in the context of carbohydrate content of the diet.) Without appropriate referencing, statements about the contribution of MetS to mortality should be removed.

Line 61-62 – studies of tech-mediated interventions in participants at risk of MetS – citations 16, 25-29 – this is somewhat misleading as not all of the studies were focused on MetS e.g. Ref 25 included a wide range of people with cardiometabolic risk, Ref 26 is a systematic review and meta-analysis of people with diabetes, cardiac disease or COPD (not MetS, and not focused on prevention of MetS)

Line 73 – the meaning of the new sentence is unclear. If the authors intend to examine studies of both prevention and treatment of MetS, the title should be amended to clearly reflect that. In addition, a clear statement incorporating PICO should be developed so that the reader understands the scope of the review, in particular the population of interest. The authors have clearly made the point that young adults (define age range?) should be targeted for prevention actions; if this review is to include studies not only of young adults but also older adults, as is clearly the case, then justification of including these additional studies should be provided. Furthermore, the discussion should address age differences in the studies included in regards to the effectiveness of the interventions.

Line 83 – an updated search should be performed to capture publications from Feb 2018 to now (almost 3 years) because the field is advancing rapidly. Indeed, publications already included in this paper’s citations will not be included because they are too recent, even though they appear relevant.

Section 2.4 – include a subgroup analysis for studies with younger participants

Additional suggestions

Line 24 – Capitalize “these”

Line 31 – the criteria for MetS include low HDL-C

Line 34 – suggest deleting “, which estimates one in four adults” because it is redundant and also grammatically awkward

Line 52 – delete “According to” and the comma after “previous”

Line 118 – delete “original”

Line 146 – this new sentence does not make sense.

Paragraph beginning line 157 – because young adults are of particular interest, the number of studies focusing on this population should be included.

Line 363 – what is meant by “technology-mediated arbitration”?

Line 384 – needs grammatical revision

Line 389 – the two sentences beginning “Further, it is thought…”  and “Furthermore, since…” do not make sense and need revision

Author Response

Please check the file attached.

Reviewer 2 Report

Thanks for responding to first draft of comments however additional changes needed before paper can be accepted for publication.

Please address the highlighted text and comments as in the attached PDF document.

Overall an additional subgroup analysis (those with established MetS and those at risk of MetS) and GRADE summary of findings with revision for level of evidence needed in addition to minor spelling and grammatical errors.

Author Response

Please check the file attached.
